# Unilateral Exercise and Bilateral Vascular Health in Female Tennis Players and Active Controls

**DOI:** 10.3390/sports13040107

**Published:** 2025-04-01

**Authors:** Chanhtel E. Thongphok, Abena O. Gyampo, Elisa Fioraso, Anneli O. Ramolins, Elianna G. Hills, Claire E. Coates, Stephen J. Ives

**Affiliations:** 1Health and Human Physiological Sciences Department, Skidmore College, Saratoga Springs, NY 12866, USA; cthongph@skidmore.edu (C.E.T.); agyampo@skidmore.edu (A.O.G.); elisa.fioraso@studenti.univr.it (E.F.); aramolin@skidmore.edu (A.O.R.); ehills@skidmore.edu (E.G.H.); ccoates@skidmore.edu (C.E.C.); 2Department of Biomedicine and Movement Sciences, University of Verona, 37129 Verona, Italy

**Keywords:** central blood pressure, augmentation index, pulse wave analysis, exercise, women

## Abstract

Blood pressure (BP), interarm differences (IAD) in BP, and arterial stiffness (AS) are related to cardiovascular disease risk and are attenuated by exercise training. While active, tennis players (TP) experience bilateral differences in shear stress, and thus vascular function due to the unilateral nature of the sport. However, it is unknown if this translates into attenuated bilateral differences in peripheral blood pressure (pBP), estimated central blood pressure (cBP), and AS, which could provide insight into the local versus systemic effects of exercise training on BP in women. Purpose: to evaluate bilateral differences in pBP, cBP, and AS in Division III female college TP and healthy recreationally active (RA) age- and sex-matched controls. Methods: In a parallel design, TP (n = 10) and RA controls (n = 10) were assessed for anthropometrics, body composition, and bilateral BP measurements using oscillometric cuff technique. Results: TP and RA were well-matched for body weight, body fat percentage, and BMI (all, *p* > 0.69). Interaction of arm and group, and effects of arm, or group were insignificant for pSBP and pDBP (all, *p* > 0.137). IAD in pSBP tended lower in TP (*p* = 0.096, d = 0.8), but IAD in cSBP was lower (*p* = 0.040, d = 0.8). Augmentation pressure and index were different between arms (*p* = 0.02), but no interactions (group by arm) were observed (*p* > 0.05). Conclusions: In groups well-matched for age and body composition, TP tended to have lower BP and IAD in pSBP, but cSBP revealed ~50% lower IAD in TP. Thus, measurement site and exercise training matter when assessing arterial stiffness and interarm differences in BP.

## 1. Introduction

Globally, cardiovascular disease (CVD) is the leading cause of mortality and has been attributed to roughly 18 million deaths per year [1]. The most common cardiovascular diseases contributing to the fatality rate include coronary heart disease, stroke, and rheumatic heart disease [1,2]. According to the National Health and Nutrition Examination Survey, approximately 70% of major CVD events in the United States resulted from low cardiovascular health (CVH) but could have been prevented by attaining a high level of CVH [3]. Maintaining a healthy diet, performing regular exercise, the cessation of tobacco use, limiting alcohol consumption, and, ultimately, controlling blood pressure (BP) can significantly decrease CVD risk [1,3]. Understanding how factors such as exercise alter BP and thus CVD risk is worthy of further study.

The vascular system primarily consists of the arteries and veins that carry blood throughout the body and supply tissues with oxygen and nutrients [4]. With aging or disease, arteries tend to stiffen, decreasing the ability of arteries to dilate, likely due to the development of fibrosis and collagen cross-linked products in the arterial wall [5,6,7]. The aorta and the central arteries are complex components within the vascular system that buffer oscillations in blood flow and blood pressure via elasticity [8]. A reduction in buffering function results in increased left ventricular afterload and myocardial oxygen demand, and a decrease in coronary blood flow, which can lead to myocardial hypertrophy and ischemia [6,8]. Arterial stiffness has also been associated with impaired arterial baroreflex regulation, which signifies a collective negative effect on the heart, arteries, and the autonomic nervous system [8]. In addition, arterial stiffness has been directly related to hypertension, hyperlipidemia, elevated BMI, smoking, and diabetes, which are common CVD risk factors [7]. Thus, vascular stiffness may be a valuable predictor of cardiovascular-related morbidity and mortality and should be adopted clinically [7].

Past studies have assessed the cardiovascular benefits of exercise training, but some questions remain about whether these effects are locally or systemically mediated [9]. To better parse out these local vs. systemic effects of exercise on vascular health, past studies have used unilateral activities such as racquet sports [9,10,11]. Sinoway and colleagues [11], may have been the first to explore whether exercise exerts a systemic or local effect on vascular health, using habitual tennis players, revealing that TP had 42% difference in reactive hyperemia between arms whereas fitness-matched controls did not. Other studies have revealed similar findings that vascular function is enhanced in the dominant arm of racquet sport athletes, suggesting that exercise impacts the vasculature in a localized fashion [9,10,11]. However, these prior studies of unilateral activity were conducted exclusively in men. Further, interarm differences (IAD) in blood pressure have been widely investigated [12], revealing that a difference of more than 10 mmHg in systolic BP between the two arms is linked with an increase in coronary artery disease [12,13,14,15,16]. However, it has yet to be determined if bilateral differences in training such as with tennis manifest in the assessment of peripheral blood pressure, derived central blood pressure, and arterial stiffness, especially in women who have not been studied in this regard.

Therefore, the main purpose of this study was to evaluate the bilateral differences in peripheral and central blood pressure and arterial stiffness in women. Second, using female tennis players (TP) as a model of large interarm training differences, we compared them to recreationally active matched controls (RA). It was hypothesized that the TP would have lower BP and arterial stiffness than RA-matched controls, which would be lower in the dominant arm, and more so in the TP, suggesting greater interarm differences in blood pressure. Understanding the potential role of training in IAD in BP can help elucidate contributing factors and thus the clinical interpretation of IAD in BP.

## 2. Methods

### 2.1. Subjects and General Procedures

Twenty healthy college-aged females (10 TP, 10 RA) aged 18–23 years old were recruited for this study via word of mouth on the Skidmore College campus. In a matched pairs design, using a one-tail approach, a large effect size, alpha of 0.05, to achieve a minimum power of 0.8, a total of 12 participants would be needed. To be included, if TPs were official members of the Skidmore College women’s tennis team and performed 5–7 days a week of aerobic exercise for a minimum of 1 h per session, they met the criteria for inclusion. RA controls were described as individuals who performed aerobic exercise at least 2–3 days a week for a minimum of 30 min per session but had no significant history of participation in racquet sports. Exercise habits were captured via self-report in an online survey. Subjects were excluded from this study if they had any chronic disease(s), had an injury that would prevent exercise training, or were regular tobacco smokers. Matching was carried out on an individual basis to ensure the groups were not different from each other in terms of general characteristics. Preparatory to testing sessions, subjects were asked to consume their normal daily meals but to avoid eating within the 4 h preceding their visit, and were not permitted to consume caffeine for a minimum of 6 h prior to data collection or perform moderate to vigorous physical activity for at least 12 h before data collection. Participants were encouraged to drink 2 glasses of water within 2 h of their visit to ensure euhydration for the assessment of body composition. The women were studied, according to self-report, during the menses or early follicular phase of the menstrual cycle (day 0–7) or the placebo phase of a chemical contraceptive, despite a growing argument that it is not necessary [17]. All participants provided written informed consent prior to participation. The study was reviewed and approved by the local Institutional Review Board (IRB#2301-1071) and was conducted in accordance with the most recent revisions to the Declaration of Helsinki.

### 2.2. Procedures

Participants who were enrolled were asked to come into the laboratory for a single-visit cross-sectional study (Figure 1). All visits were performed in the afternoon between 13:00–17:00 to control for diurnal variation. Upon arrival at the laboratory, anthropometric measurements were obtained to characterize participants and allow for matching on these parameters. Height was obtained using a stadiometer (Seca, Mt. Pleasant, SC, USA), while weight body weight and composition were obtained using a bioelectric impedance analysis body composition scale (RD-545, Tanita, Arlington Heights, IL, USA). The Tanita BIA has been documented to be a reliable and valid approach [18], and we have documented that in-house testing yields a within-day average coefficient of variation of 0.4% across 3 trials [19].

Participants were then asked to lie supine in a quiet dimly lit room while they were fitted with a blood pressure cuff on each arm to minimize disruption of measurement between arms. Appropriate cuff size was confirmed using mid-bicep circumference measurements using a Gulick tape measure. In a counterbalanced fashion, after 10 min of rest in the supine position, blood pressure readings were obtained using the oscillometric cuff technique (SphygmoCor Xcel, Atcor, Naperville, IL, USA), in duplicate, and then averaged. Central pressures were derived from the brachial pressure waveforms using a generalized transfer function, with subsequent pulse wave analysis for the determination of factors related to arterial stiffness; specifically, central pulse pressure, augmentation pressure, and augmentation index. The SphygmoCor Xcel device has been validated against invasive measures of blood pressure [20] and waveform analysis [21]. Interarm differences (IAD) were calculated individually, on an absolute basis, for both central and peripheral systolic and diastolic blood pressures. The testing order of arms (dominant vs. non-dominant) was counterbalanced at the participant level, by having even participant ID#s having their dominant measured first and non-dominant measured second, while odd ID#s were tested in the opposite order. The time between measurements or arms was standardized at roughly 2 min apart. 

### 2.3. Statistical Analysis

Descriptive and inferential statistics for all data were computed and reported using open-source software (JASP v0.18.1, University of Amsterdam, Amsterdam, The Netherlands). To test the interaction of arm and group on vascular stiffness and blood pressure between female TPs and Ras, a two-way mixed measure analysis of variance (ANOVA) was employed and an appropriate measure of effect size (partial eta-squared, η^2^_p_) was provided, using 0.01, 0.06, and 0.14 as small, medium, and large effects, respectively. Tests of assumption were assessed using the Shapiro–Wilk test of normality and homogeneity test that produced Levene’s equality of variances. If a significant violation was found, the Greenhouse–Geisser correction was applied to adjust the degrees of freedom. Tests of single effect (e.g., group) were conducted using independent samples *t*-test, and Cohen’s d estimate of effect size appropriate was included to complement *p*-values, using 0.2, 0.5, and 0.8 as small, medium, and large effects, respectively. Chi-squared and contingency tables were used to compare the frequency of positive IAD (≥10 mmHg) [14] on peripheral SBP of the TP and RA controls. The level of statistical significance was established at *p* < 0.05, and all data were expressed as means ± SD unless noted otherwise.

## 3. Results

### 3.1. Subject Characteristics of TP and RA Controls

The tennis players and recreationally active controls were well-matched based on age, height, weight, body mass index, and body composition (all *p* > 0.05, Table 1). Further, while the testing order was counterbalanced, we tested for order effect (1st vs. 2nd measurement) and found no such effects (all, *p* > 0.05).

### 3.2. Blood Pressure and Interarm Differences Between TP and RA Controls

First, resting heart rate (HR) was not different between arms (73 ± 12 vs. 71 ± 11 beats/min, dominant vs. non-dominant, respectively, *p* = 0.363, d = 0.21). In peripheral systolic blood pressure, there was no significant interaction of arm and group (*p* = 0.872, η^2^_p_ = 0.002), no main effect of arm (*p* = 0.137, η^2^_p_ = 0.125), nor a main effect of group (*p* = 0.137, η^2^_p_ = 0.125) (Figure 2A). For peripheral diastolic blood pressure, there was not an interaction of arm and group (*p* = 0.428, η^2^_p_ = 0.037), arm (*p* = 0.762, η^2^_p_ = 0.006) or for group (*p* = 0.141, η^2^_p_ = 0.123) (Figure 2B). The interarm difference in peripheral systolic blood pressure was not significantly different between groups, although TP exhibited a large effect with lower IAD than RA controls (*p* = 0.096, d = 0.8, Figure 2C). The interarm difference in peripheral diastolic blood pressure was not significantly different between TP and RA controls (*p* = 0.970, d = 0.0, Figure 2D). In terms of the prevalence of IAD (≥10 mmHg) in peripheral SBP, only 1 of 10 TP were positive for IAD, while 5 of 10 RA controls were positive for IAD, and this approached statistical significance (χ^2^ = 3.3, *p* = 0.069).

In terms of central SBP, there was no significant interaction of arm and group (*p* = 0.929, η^2^_p_ = 0.000), no effect of arm (*p* = 0.656, η^2^_p_ = 0.125), but a tendency for group effect with TP having lower cSBP than RA controls (*p* = 0.071, η^2^_p_ = 0.179, Figure 3A). In central DBP (Figure 3B) there was not a significant interaction observed for arm and group (*p* = 0.796, η^2^_p_ = 0.004), no main effect of arm (*p* = 0.796, η^2^_p_ = 0.004) and no effect of group (*p* = 0.117, η^2^_p_ = 0.138). For mean arterial pressure (MAP), there was no significant interaction of arm and group (*p* = 1.000, η^2^_p_ = 0.000), no effect of arm (*p* = 0.594, η^2^_p_ = 0.017), but MAP tended to be lower (*p* = 0.061, η^2^_p_ = 0.191) in the TP vs. RA controls (85 ± 8 vs. 92 ± 8 mmHg). The interarm difference in central systolic blood pressure was significantly different between groups (*p* = 0.040, d = 1.0, Figure 3C), with TP exhibiting a lower IAD than RA controls. The interarm difference in central diastolic blood pressure was not different between groups (*p* = 0.42, d = 0.4, Figure 3D).

### 3.3. The Role of Arm in Pulse Wave Analysis Estimates of Arterial Stiffness of TP and RA Controls

There was no significant interaction of arm and group on central pulse pressure (*p* = 0.802, η^2^_p_ = 0.004), no effect of arm (*p* = 0.802, η^2^_p_ = 0.004), and no effect of group (*p* = 0.427, η^2^_p_ = 0.037, Figure 4A). For augmentation pressure, there was no significant interaction of arm and group (*p* = 0.766, η^2^_p_ = 0.005), and no effect of group (*p* = 0.276, η^2^_p_ = 0.069), but there was a significant arm effect (*p* = 0.024, η^2^_p_ = 0.264, Figure 4B) with the dominant arm having lower augmentation pressure. Similarly in the augmentation index, there also was no significant interaction of arm and group (*p* = 0.484, η^2^_p_ = 0.029), and no effect of group (*p* = 0.201, η^2^_p_ = 0.094), but there was a significant arm effect (*p* = 0.027, η^2^_p_ = 0.257, Figure 4C) with the dominant arm having lower augmentation index. As there were no interactions of arm and group (*p* = 0.120, η^2^_p_ = 0.136) or main effects for arm (*p* = 0.304, η^2^_p_ = 0.062) or group (*p* = 0.386, η^2^_p_ = 0.044) observed in heart rate, the augmentation index normalized to 75 beats/min reflected the same conclusions as the unadjusted AIx.

## 4. Discussion

This study aimed to evaluate the bilateral differences in central and peripheral blood pressure and arterial stiffness in Division III female college tennis players and recreationally active female controls who were matched for age, height, weight, and body composition. The current findings noted medium to large effect sizes in BP for group, suggesting that both peripheral and central blood pressures tended to be lower in the TP, perhaps as a result of greater exercise training intensity and/or volume. The IAD in peripheral systolic blood pressure tended to be lower in TP (large effect size), while IAD in central systolic blood pressure was lower (large effect size), and both were similar in magnitude (~50%). In terms of measures of arterial stiffness, specifically central pulse pressure, augmentation pressure, and index all tended to be lower in TP with medium to large effect sizes, but a significant arm difference was revealed with lower AP and AIx in the dominant arm, irrespective of group, as we found no significant interaction effects of arm and group. These findings are somewhat contrary to our initial hypothesis in that we had expected the TP to exhibit greater bilateral differences or IAD in blood pressure. Collectively, the current study highlights that assessment of both peripheral and central blood pressures may be needed to elucidate differences between groups in IAD in BP, whether greater training in TP likely reduces IAD in BP of TP, and that the arm that is measured matters in terms of augmentation pressure and index.

### 4.1. Interarm Differences in Blood Pressure

Interarm differences (IAD) in blood pressure have been widely investigated [12,13,14,15,16] over the years and many studies have highlighted that a difference of more than 10 mmHg in systolic blood pressure between the two arms is linked with an increase in coronary artery disease [12,13,16]. Given that the IAD is an important parameter to assess cardiovascular risk, the method of measurement should be as accurate as possible. Usually, during clinical or screening visits, the BP is measured on a single arm, assumedly the one closest to the clinician or wall-mounted sphygmomanometer, without assessing arm dominance or bilateral measurements. Previous literature has consistently documented significant differences between the two arms, even in preclinical populations [14,15], but as far as we know there are no studies that have investigated IAD in BP (peripheral or central) that considered the influence of unilateral training. To obtain a better estimation of cardiovascular risk, blood pressure should be taken in both arms, dominant and non-dominant, as the American Heart Association suggests [22], and the higher reading arm used for further monitoring.

In the current study of young healthy active women, we did not find statistically significant main effects for arm in BP, whether peripheral or central, systolic or diastolic. Though we did find differences in the IAD between TP and RA controls in the central systolic BP with a similar trend in peripheral systolic BP, in both cases the IAD was ~50% lower in the TP. However, we did not find either the TP or RA controls, on average, met the threshold of 10 mmHg, on either peripheral or central systolic BP [12,13,14,15,16], although the proportion of those positive for IAD tended to be greater in the RA-matched controls. Even when combining TP and RA controls, the average peripheral systolic BP IAD was below the 10 mmHg threshold. Thus, some inherent IAD in BP is to be expected, but age, exercise training status or fitness, and sex, especially in pre-menopausal women, likely play a role in the lesser prevalence or magnitude of the IAD. In agreement with Kim et al., who investigated IAD in ambulatory patients without CVD, we found no significant differences between arms, and that the IAD was lower than the critical cutoff of 10 mmHg on systolic [14]. The current findings are somewhat in contrast to the previous studies on the local vs. systemic effects of exercise training on vascular function using racquet sport athletes, which, as mentioned previously, focused exclusively on men [9,10,11]. If the unilateral nature of tennis were to elicit greater vasodilation in the dominant arm over the non-dominant arm, and thus potentially lower vascular resistance and blood pressure, it did not manifest in the measurement of peripheral or estimated central blood pressures. In fact, the interarm difference in BP was attenuated in the TP, which might be due to greater aerobic fitness in the TP, or greater two-handed tennis play, especially in women’s tennis (e.g., two-handed backhand). This novel finding should be followed up. More research with larger sample sizes, in both sexes, and across the menstrual cycle in women is needed on IAD in peripheral and central BP to determine the potential relation to CVD risk.

### 4.2. Bilateral Differences and Training Level

In the present study, we investigated the interarm difference in tennis players, expecting that the athletes, more so than RA-matched controls, would have the dominant arm performing much more than the other. Physical activity (PA) affects vascular function, in that higher levels of PA improves vascular health [23]. Interestingly, the results of this study suggest that the augmentation index was better in the dominant arm seemingly irrespective of group, as there were no significant differences between the groups; however, it should be noted that medium effect sizes were noted with TP having lower indices of arterial stiffness. By recruiting tennis players, we wanted to compare differences between the dominant and non-dominant arm and understand how it may or may not manifest in measurements derived from blood pressure. Specifically, large artery stiffness as assessed by central pulse pressure, augmentation pressure, and augmentation index, are all useful to assess cardiovascular health [6,7,8,23] and they are also linked with all-cause mortality [24]. However, there is a paucity of data comparing these parameters between arms for individuals and whether unilateral or dominant arm training augments or attenuates the responses as compared to controls. Prior studies have investigated the systemic vs. local effects of exercise on vascular health using racquet sport athletes [9,10,11], with most of them specifically investigating the effect in tennis players [10,11]. From these studies, reduced wall thickness, arterial remodeling, peripheral vascular adaptation, and an increase in vasodilation capacity are some of the local consequences of exercise on the vascular system, at least in men [9,10,11]. Congruous with the generalized benefits of exercise training on arterial stiffness [6,8], the tennis players exhibited medium effect sizes for lower indicators of large artery stiffness, but a greater interarm effect was not observed in the TP. This novel finding might suggest sex specificity in the vascular adaptation to unilateral training, or simply differences in the mechanisms underlying the assessments, or a more bilateral nature of the sport in women. Interestingly, there was a general or main effect of arm, with the dominant arm in both groups exhibiting significantly lower augmentation pressure and index. Again, this new information suggests that habitual dominant arm use can positively affect the assessment of augmentation pressure and index, and should be considered when making such measurements. As far as we know, the literature is sparse on interarm differences in the oscillometric cuff technique with subsequent generalized transfer function analysis of large artery stiffness and whether enhanced training unilaterally adversely or positively influences these parameters. More research on interarm differences in pulse wave analysis is needed in larger samples of men and women, leveraging training and/or disuse models.

### 4.3. Experimental Considerations

In the current study, while embracing a repeated measures factor (within-subject examination of bilateral differences in BP and arterial stiffness), and with careful matching of RA controls to TP, the study was carried out with strengths and limitations. As a strength, we did control for the menstrual cycle phase to minimize the potential effects of hormonal variation, despite a growing view that such control is unnecessary [17]. Regarding limitations, while the numbers were relatively low (20 total participants), focusing on women in a single sport and status (Division III collegiate), limiting the number of total available participants, the entire women’s tennis team was recruited and studied, with equal numbers of matched controls. In a matched pairs design, using a one-tail approach, a large effect size, alpha of 0.05, to achieve a minimum power of 0.8, a total of 12 participants would be needed, which we exceeded. Recruiting TP outside of the institution would have likely introduced greater heterogeneity in the data (i.e., different sex, age groups, experience, coaching/training programs, etc.) and thus was avoided. Finally, there was no formal assessment of aerobic capacity (e.g., VO_2_peak) or muscle strength (e.g., handgrip strength) of either group and this could have helped to elucidate the degree to which fitness may or may not have played a role in the current findings. Given the closeness of BMI and body fat percentage, it is tempting to speculate the TP and matched RA controls had a similar level of fitness, which is tenable given recent reports of high-level female tennis players’ VO_2_max of 40 mL/kg/min being somewhat similar to that of active college-age females [25]. However, training volume and/or history may have also played a role and should be considered in future studies, along with the assessment of aerobic fitness (e.g., VO_2_max). Finally, future studies should assess the same metrics used in the prior studies of male racquet sports athletes and explore potential bilateral differences in vascular structure (e.g., brachial artery diameter) and function (e.g., flow-mediated dilation and/or reactive hyperemia) in women, and perhaps other sports (e.g., pickleball).

## 5. Conclusions

Although previous research has investigated bilateral differences in vascular health in athletes with expected training-induced localized effects (e.g., tennis players, squash, etc.), the results of this study extend these findings for the first time to women and bilateral differences in BP, central BP, and arterial stiffness. Unlike the accentuated local vascular adaptations that have been noted to be enhanced in the dominant arm of racquet sport athletes, we observed a diminution of the interarm differences in systolic blood pressure as compared to age-, sex-, BMI-, and body fat-matched controls. In terms of augmentation pressure and index, studying the dominant vs. non-dominant arm affects the magnitude and is thus a crucial consideration when designing and conducting assessments of cardiovascular health. Collectively, it may be that a higher degree of fitness in TP, perhaps independent of bilateral training, actually reduces IAD in BP and thus CVD risk. Clinically, the present study highlights the importance of assessing BP bilaterally and assessing exercise history as prudent measures in assessing cardiovascular health profile.

## Figures and Tables

**Figure 1 sports-13-00107-f001:**
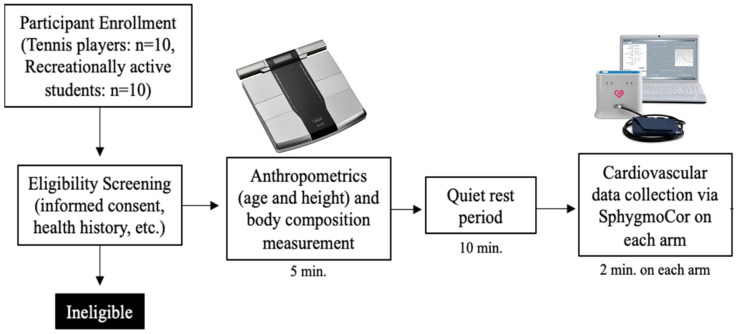
Experimental overview. Note: the dominant and non-dominant arms were counterbalanced.

**Figure 2 sports-13-00107-f002:**
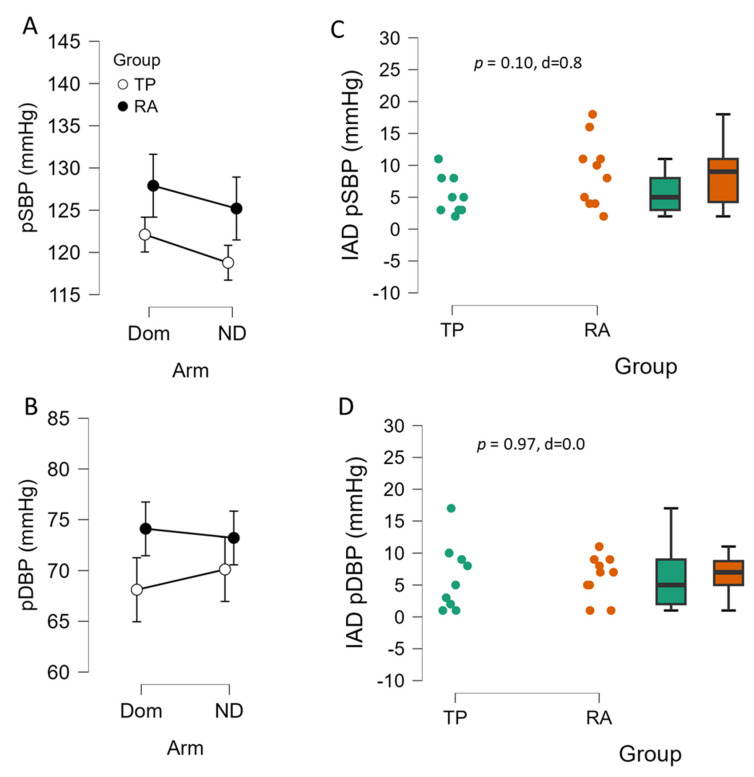
Bilateral difference in peripheral systolic blood pressure (pSBP, (**A**)) and diastolic blood pressure (pDBP, (**B**)) and interarm difference (IAD, (**C**,**D**)) of female college tennis players (TP, n = 9) vs. recreationally active (RA, n = 10) female controls. Data are expressed as means ± 95% CI, and as individual plots with box-whisker. Panel (**C**,**D**) independent samples *t*-test *p* value and Cohen’s d.

**Figure 3 sports-13-00107-f003:**
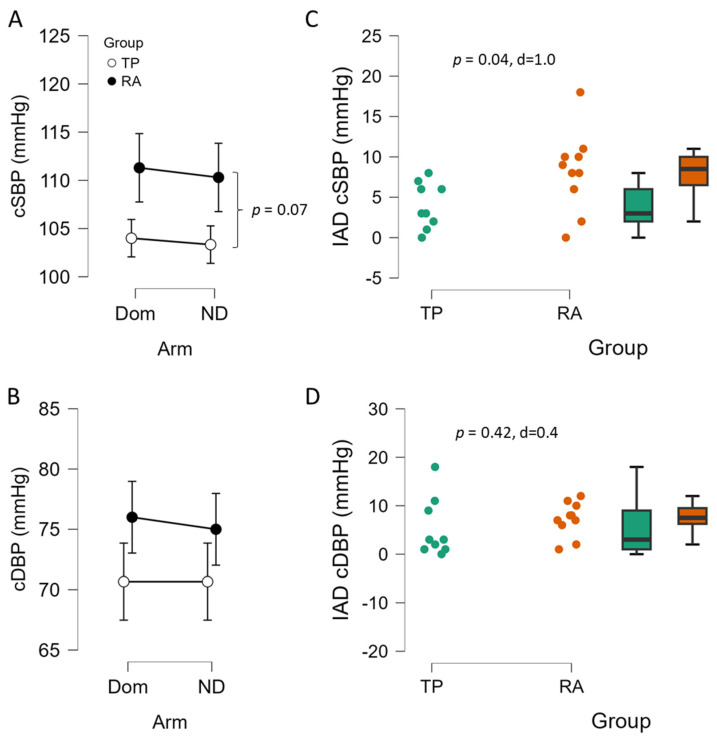
Bilateral difference in central systolic blood pressure (cSBP, (**A**)) and diastolic blood pressure (cDBP, (**B**)) and interarm difference (IAD, (**C**,**D**)) of female college tennis players (TP, n = 9) vs. recreationally active (RA, n = 10) female controls. Data are expressed as means ± 95% CI, and as individual plots with box-whisker. Panel A, main effect of group *p* = 0.071, η^2^_p_ = 0.179. Panel (**C**,**D**) independent samples *t*-test *p* value and Cohen’s d.

**Figure 4 sports-13-00107-f004:**
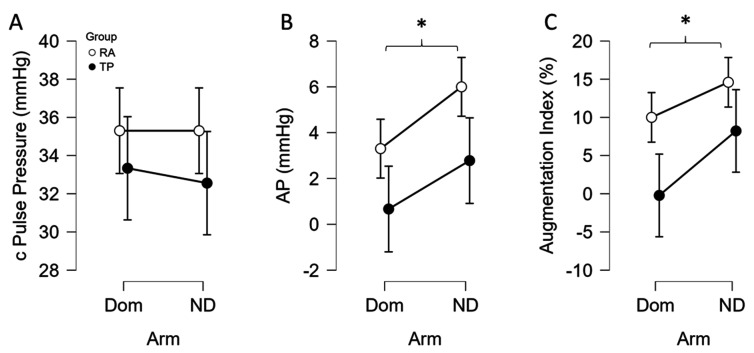
Bilateral differences of dominant (Dom) and non-dominant (ND) arm in central pulse pressure (cPP, (**A**)), aortic augmentation pressure (AP) (**B**), and augmentation index (**C**) of female college tennis players (TP, n = 9) vs. recreationally active (RA, n = 10) female controls. Data are expressed as means ± 95% CI. * *p* < 0.05 main effect of arm, dominant vs. non-dominant arm.

**Table 1 sports-13-00107-t001:** Subject characteristics.

Characteristics	TP (n = 10)	RA (n = 10)	Sig
Age (years)	19.4 ± 1.5	21.2 ± 1.1	*p* = 0.750
Height (cm)	163.9 ± 5.1	166.1 ± 5.3	*p* = 0.520
Weight (kg)	62.4 ± 11.5	62.1 ± 14.4	*p* = 0.987
BMI (kg/m^2^)	21.9 ± 2.0	21.3 ± 4.3	*p* = 0.860
Body fat (%)	22.6 ± 6.9	23.4 ± 8.0	*p* = 0.434
Heart rate (beats/min)	70 ± 10.5	74 ±12.4	*p* = 0.422

## Data Availability

The data are available upon reasonable request to the corresponding author. The data are not publicly available due to privacy reasons.

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
