# Peer review of "Unilateral Exercise and Bilateral Vascular Health in Female Tennis Players and Active Controls"

_sports, 2025, doi:10.3390/sports13040107_

Round 1

Reviewer 1 Report

Comments and Suggestions for Authors

After a thorough review of your manuscript, we appreciate the novelty and scientific rigor of your study on the effects of unilateral exercise on bilateral vascular health in female tennis players. The research is well-structured, methodologically sound, and contributes valuable insights to the field. However, we have identified several areas where minor revisions are required to enhance the clarity and impact of your work.
1. Your hypothesis suggests that tennis players (TP) would exhibit greater inter-arm differences (IAD) due to unilateral training. However, your findings indicate the opposite (lower IAD in TP). Please provide a clearer explanation for this unexpected outcome in the discussion section.

2. Sample size is small, please acknowledge this limitation in greater detail in the discussion.

3. While the study controlled for menstrual cycle phase, other factors like dietary intake, hydration status, and prior physical activity could influence blood pressure measurements. Please revise it.

4. Some figures have overlapping data points, making interpretation difficult. Please consider adjusting visualization techniques for better clarity.

5. In Table 1, include p-values for group comparisons to explicitly show statistical significance (or lack thereof) between groups.

Author Response

After a thorough review of your manuscript, we appreciate the novelty and scientific rigor of your study on the effects of unilateral exercise on bilateral vascular health in female tennis players. The research is well-structured, methodologically sound, and contributes valuable insights to the field. However, we have identified several areas where minor revisions are required to enhance the clarity and impact of your work.

RESPONSE: We sincerely appreciate the time and effort of the reviewer in helping to improve the current manuscript, as well as the recognition of the quality of our work. We have responded to the comments below in a point-by-point fashion below, highlighting changes made to the manuscript, where appropriate.
1. Your hypothesis suggests that tennis players (TP) would exhibit greater inter-arm differences (IAD) due to unilateral training. However, your findings indicate the opposite (lower IAD in TP). Please provide a clearer explanation for this unexpected outcome in the discussion section.

RESPONSE: Indeed, our hypothesis was informed by the literature that found greater vascular function in the preferred arm of tennis players (e.g., PMID: 8872666), suggesting inter-arm differences in vascular health, as assessed by peripheral vascular function. Thus, it was reasonable to assume that we might find greater interarm differences in the TP vs. controls. Somewhat to our surprise, the TP actually had lower interarm differences in the current suite of assessments, but this does agree, in hindsight with lower risk of CVD in this active group and perhaps lower IAD in BP in this population is in part explanatory. We have revised the discussion section in response to this and the other reviewers comment.  

  1. Sample size is small, please acknowledge this limitation in greater detail in the discussion.

RESPONSE: Agreed, the sample size is relatively small. We have increased the level to which we address the sample size estimation in the methods, and again in the discussion (new lines 316-318 as well as 312-316, and 318-320). As now noted in the manuscript, we recruited the entire women’s team, which more than met our sample size estimation for adequate statistical power. Further, for homogeneity purposes and logistical issues, we stuck with collegiate women from a single team. This is now discussed in the manuscript, thank you for the opportunity to clarify.

  1. While the study controlled for menstrual cycle phase, other factors like dietary intake, hydration status, and prior physical activity could influence blood pressure measurements. Please revise it.

RESPONSE: We appreciate the reviewer bringing this up, this is addressed in the methods line 87-93. We appreciate the opportunity to clarify as these controls were in fact part of the design.

  1. Some figures have overlapping data points, making interpretation difficult. Please consider adjusting visualization techniques for better clarity.

RESPONSE: This is an interesting point to consider. The reviewer is correct, in that some of the data points are overlapping, that in and of itself is telling in interpretation. If there is little separation that suggests homogeneity within or between groups. We believe that these figures are already visually optimized balancing showing visual spread with reasonable scaling of the y axis within the spread of the data.

  1. In Table 1, include p-values for group comparisons to explicitly show statistical significance (or lack thereof) between groups.

RESPONSE: Thank you for this suggestion, we have since added this information to the table.

Reviewer 2 Report

Comments and Suggestions for Authors

General Comments

The article investigates the impact of unilateral exercise on bilateral vascular health by comparing female tennis players (TP) to recreationally active (RA) female controls. It focuses on bilateral differences in peripheral blood pressure (pBP), central blood pressure (cBP), and arterial stiffness (AS) measures such as augmentation pressure (AP) and augmentation index (AIx). The study finds that despite the unilateral nature of tennis, TP exhibit lower inter-arm differences (IAD) in blood pressure—particularly central systolic blood pressure (cSBP)—compared to RA controls, suggesting that overall exercise training may reduce IAD rather than amplify it as hypothesized. Additionally, AS measures like AP and AIx were lower in the dominant arm for both groups, indicating an arm-specific effect independent of training status. This research provides valuable insights into how exercise training influences vascular health and highlights the importance of measurement site in assessing cardiovascular risk factors.

Specific Comments

Introduction

The introduction offers a thorough overview of cardiovascular disease (CVD) and vascular health but could be more concise. Consider reducing general CVD statistics and focusing on the research gap—specifically, the lack of studies on bilateral vascular differences in women and the potential effects of unilateral exercise. This will sharpen the focus and make the introduction more engaging.

Methods:

  • The methods note that recreationally active (RA) controls were matched for age, height, weight, BMI, and body fat percentage, but it’s unclear whether this was individual or group matching. Specify the process for transparency. Also, clarify how exercise habits were verified (e.g., self-report, activity logs) to ensure consistency.
  • Controlling for menstrual cycle phase is a strength, but indicate how this was confirmed (e.g., self-report, hormone levels). If self-reported, acknowledge potential inaccuracies. Additionally, state whether participants with irregular cycles were excluded, as this could influence vascular measurements.
  • Counterbalancing arm testing order is a good practice, but confirm whether the order affected blood pressure (BP) readings. A brief statement (e.g., “No order effect was observed”) would reassure readers. Also, specify the time interval between measurements on each arm for consistency.
  • The SphygmoCor Xcel is validated, but briefly note its reliability in your population (young, healthy females), as validation studies often focus on older or clinical groups.

Results

  • The chi-squared test for IAD prevalence (p=0.069) is near significance. Report exact frequencies (e.g., “1 of 10 TP vs. 5 of 10 RA had IAD ≥10 mmHg”) to provide clarity. This could also be presented in a table for emphasis.
  • You report no differences in heart rate (HR) between arms or groups, but including HR values in a table (e.g., Table 1) would give context for augmentation index (AIx) normalization.

Discussion

Explore Mechanisms for Unexpected Findings:

The lower IAD in TP contradicts your hypothesis and prior studies in men. Offer possible explanations, such as: Systemic aerobic fitness in TP reducing overall vascular stiffness, minimizing IAD. Non-dominant arm activity in TP (e.g., two-handed backhands) reducing bilateral differences. Sex-specific vascular responses differing from men.

Compare to Male Studies:

Contrast your findings with research on male racquet sport athletes (e.g., Sinoway et al., 1986; Green et al., 1996). Highlight whether reduced IAD in women is novel and discuss potential sex differences in vascular adaptation.

Discuss Arm-Specific Effects on Arterial Stiffness:

The significant arm effect for augmentation pressure (AP) and AIx (lower in the dominant arm) is notable. Suggest explanations, like greater habitual use of the dominant arm in both groups, to contextualize why this wasn’t specific to TP.

Acknowledge Fitness and Training Volume:

The conclusion links greater training in TP to reduced IAD, but training volume or intensity wasn’t quantified beyond frequency. Rephrase to “the training status of TP may reduce IAD” and recommend measuring VOâ‚‚max or training load in future studies.

Expand on Practical Implications:

Discuss how findings might impact clinical practice, such as measuring BP in both arms or considering exercise history when assessing IAD.

Suggest Future Research Directions:

Beyond larger samples, propose studies to explore mechanisms (e.g., vascular imaging for structural differences) or IAD across sports or training types.

Author Response

The article investigates the impact of unilateral exercise on bilateral vascular health by comparing female tennis players (TP) to recreationally active (RA) female controls. It focuses on bilateral differences in peripheral blood pressure (pBP), central blood pressure (cBP), and arterial stiffness (AS) measures such as augmentation pressure (AP) and augmentation index (AIx). The study finds that despite the unilateral nature of tennis, TP exhibit lower inter-arm differences (IAD) in blood pressure—particularly central systolic blood pressure (cSBP)—compared to RA controls, suggesting that overall exercise training may reduce IAD rather than amplify it as hypothesized. Additionally, AS measures like AP and AIx were lower in the dominant arm for both groups, indicating an arm-specific effect independent of training status. This research provides valuable insights into how exercise training influences vascular health and highlights the importance of measurement site in assessing cardiovascular risk factors.

RESPONSE: We genuinely appreciate the time and effort of the reviewer in helping to improve the current manuscript, as well as the recognition of the quality of our work. We have responded to the comments below in a point-by-point fashion below, highlighting changes made to the manuscript, where appropriate.

Specific Comments

Introduction

The introduction offers a thorough overview of cardiovascular disease (CVD) and vascular health but could be more concise. Consider reducing general CVD statistics and focusing on the research gap—specifically, the lack of studies on bilateral vascular differences in women and the potential effects of unilateral exercise. This will sharpen the focus and make the introduction more engaging.

RESPONSE: We have edited the introduction to better highlight the gaps in the literature. However, regarding the general statistics, we feel this is relevant for broader public health and screening for CVD. A review (PMID: 27977471) revealed that inter-arm differences in BP is one of 29 potential sources of error in BP measurement. Given measurement of BP is the key metric used in the detection and management of hypertension, thus we feel it is relevant to discuss. That said we have made targeted edits to better link this discussion to the topic at hand.

Methods:

  • The methods note that recreationally active (RA) controls were matched for age, height, weight, BMI, and body fat percentage, but it’s unclear whether this was individual or group matching. Specify the process for transparency. Also, clarify how exercise habits were verified (e.g., self-report, activity logs) to ensure consistency.

RESPONSE: This is an interesting question, it is a little bit of both, we try to match on an individual level, so that the groups are similar and thus not different. As for exercise habits this was done via self-report and is now noted in the methods Line 88-89.

  • Controlling for menstrual cycle phase is a strength, but indicate how this was confirmed (e.g., self-report, hormone levels). If self-reported, acknowledge potential inaccuracies. Additionally, state whether participants with irregular cycles were excluded, as this could influence vascular measurements.

RESPONSE: Thank you for the opportunity to clarify this, menstrual cycle phase was done by self-report. We have now clarified this in the manuscript.

  • Counterbalancing arm testing order is a good practice, but confirm whether the order affected blood pressure (BP) readings. A brief statement (e.g., “No order effect was observed”) would reassure readers. Also, specify the time interval between measurements on each arm for consistency.

RESPONSE: We appreciate the reviewer bringing this up. We did document testing order and tested for it and found no evidence of an order effect. We have also included the information about time between arms in the methods (lines 132-133).

  • The SphygmoCor Xcel is validated, but briefly note its reliability in your population (young, healthy females), as validation studies often focus on older or clinical groups.

RESPONSE: This is an interesting point, the reason why the Sphygmocor has been validated in older and/or clinical populations is because to validate against direct measures you need arterial and/or central lines with a pressure transducer. This is invasive and generally not possible outside of the cardiac catheterization lab in a hospital. That said, if one examines any of the studies that have done the correlations between invasive and the non-invasive Sphygmocor device the associations were across a wide range of pressures, including those seen in young healthy populations. Thus, while not validated in this population per se, there is a relation between the device and invasive measures across a wide range of pressures, which quells the concern regarding validation in a non-clinical population. Further, reliability is another interesting concept as blood pressure is variable and, in fact, greater short-term BPV is associated with lower CVD risk. Thus, in this younger healthier population the reliability within or between day a significant component of the variability is inherent BP variability and has less to do with the device per se.

Results

  • The chi-squared test for IAD prevalence (p=0.069) is near significance. Report exact frequencies (e.g., “1 of 10 TP vs. 5 of 10 RA had IAD ≥10 mmHg”) to provide clarity. This could also be presented in a table for emphasis.

RESPONSE: Thank you for bringing this up, this is exactly reported in the manuscript (line 169-171).

  • You report no differences in heart rate (HR) between arms or groups, but including HR values in a table (e.g., Table 1) would give context for augmentation index (AIx) normalization.

RESPONSE: We have included some of the HR values in the text of the results, specifically between arms (line 159-160). As suggested by the reviewer, we have also added the group-specific grand or marginal means (average of the two arms) to table 1.

Discussion

Explore Mechanisms for Unexpected Findings:

The lower IAD in TP contradicts your hypothesis and prior studies in men. Offer possible explanations, such as: Systemic aerobic fitness in TP reducing overall vascular stiffness, minimizing IAD. Non-dominant arm activity in TP (e.g., two-handed backhands) reducing bilateral differences. Sex-specific vascular responses differing from men.

RESPONSE: Thank you for this suggestion, we have incorporated these points into the discussion (lines 266-268).

Compare to Male Studies:

Contrast your findings with research on male racquet sport athletes (e.g., Sinoway et al., 1986; Green et al., 1996). Highlight whether reduced IAD in women is novel and discuss potential sex differences in vascular adaptation.

RESPONSE: Thank you for the opportunity to highlight this, we now include a discussion of these points in the paper (lines 294-296).

Discuss Arm-Specific Effects on Arterial Stiffness:

The significant arm effect for augmentation pressure (AP) and AIx (lower in the dominant arm) is notable. Suggest explanations, like greater habitual use of the dominant arm in both groups, to contextualize why this wasn’t specific to TP.

RESPONSE: We appreciate the reviewer's suggestion and now address these points in the discussion section of the paper (lines 298-300).

Acknowledge Fitness and Training Volume:

The conclusion links greater training in TP to reduced IAD, but training volume or intensity wasn’t quantified beyond frequency. Rephrase to “the training status of TP may reduce IAD” and recommend measuring VOâ‚‚max or training load in future studies.

RESPONSE: Thank you for this suggestion, in response to this we have made edits in line 327-328 and line 342 of the discussion section.

Expand on Practical Implications:

Discuss how findings might impact clinical practice, such as measuring BP in both arms or considering exercise history when assessing IAD.

RESPONSE: We appreciate the suggestion of the reviewer and now include this as the concluding remark of the paper (lines 343-345).

Suggest Future Research Directions:

Beyond larger samples, propose studies to explore mechanisms (e.g., vascular imaging for structural differences) or IAD across sports or training types.

RESPONSE: Thank you for this suggestion this is now included in the discussion section of the paper (lines 328-331).